# Review on Monitoring and Operation-Maintenance Technology of Far-Reaching Sea Smart Wind Farms

**Zhen Wang [1], Yaohua Guo [2,3,]\* and Haijun Wang [2,3]**

1   School of Management and Economy, Tianjin University, Tianjin 300072, China; wz010617@tju.edu.cn
2   State Key Laboratory of Hydraulic Engineering Simulation and Safety, Tianjin University, Tianjin 300072, China; bookwhj@tju.edu.cn
3   School of Civil Engineering, Tianjin University, Tianjin 300072, China
\*   Correspondence: guoyaohua@tju.edu.cn

**Abstract:** With the rapid development of global offshore wind power, the demand for offshore wind power operation and maintenance is also increasing. This paper analyzes the technology of units, monitoring of deep wind field, and operation and maintenance risks and provides an innovative direction for offshore wind power operation and maintenance. In this study, the innovation of offshore wind power operation and maintenance are discussed in regard to the aspects of operation and maintenance management, the monitoring technology of units and far-reaching wind field monitoring and risks. Combined with information technology and lean management concept, this paper evaluates the development trend and difficulties of data mining and information platforms of offshore wind power operation and maintenance. A far-reaching intelligent operation and maintenance management platform for offshore wind farms is provided and a centralized and intelligent operation and maintenance management mode is explored in order to improve the efficiency and reduce the costs. Through the research on the characteristics of 5G technology, the typical application scenarios of 5G technology in the intelligent operation and maintenance of offshore wind farms are analyzed, which provide a new solution for the efficient operation and maintenance of offshore wind farms.

**Keywords:** offshore wind power; operation and maintenance management; intelligent operation and maintenance robot; smart wind farm technology; 5G technology

## 1. Introduction

In recent years, in face of the increasing greenhouse effect and air pollution, countries around the world have accelerated the pace of exploration for clean energy and renewable energy power generation. As one of the main clean energies, wind energy, especially offshore wind energy, has developed rapidly. Compared with land wind energy, the capacity benefit of offshore wind energy resources is 20~40% higher and has the advantages of less land occupation, fast operation speed, large production power, stable operation of the system and no dust. In addition, the wind speed at sea is high, while the wind speed, wind direction change and turbulence intensity are small, which can effectively reduce the mutual wear of wind turbines and improve the service life of wind turbines. It is suitable for large-scale development in coastal areas.

Compared with land-based wind power, offshore wind turbines exist in harsh marine environments for a long period of time and the failure rate of offshore wind power is significantly higher [1]. With the large scale of global offshore wind power installation, the demand for offshore wind power operation and maintenance is also increasing. The whole life cycle of wind power operation and maintenance is becoming the focus of the offshore wind power market. On the other hand, the construction site selection of offshore wind power is developing towards deeper and farther out positions and floating wind

turbines are increasing, which also means that the operation and maintenance environment will worsen and maintenance will be more difficult. Led by European countries like the United Kingdom, many offshore wind farms have been installed and are now operational throughout the world [2]. By 2029, the total investment in the global offshore wind power operation and maintenance market will exceed $12 billion.

In the harsh environment at sea, the operation and maintenance of wind turbines are significantly affected by climate and tide. The operation requirements are high, and the effective working time is short. The ships, personnel configuration and maintenance time required to maintain different types of components are quite different. Therefore, how to select the appropriate installation mode and operation and maintenance strategy in the complex and changeable environment and limited working time at sea is of great significance when promoting the development of offshore wind power [3]. The improvement of the operation level and benefit of wind farms through digitalization and intelligence is urgent, and the construction of smart offshore wind farms has become an industry consensus. The future construction of smart offshore wind farms depends on good operation and maintenance management, which requires the support of scientific operation and maintenance strategy; intelligent fault diagnosis; monitoring technology; stable, efficient operation and maintenance vessels.

In the context of the rapid development of global offshore wind power, this paper introduces the current development situation and characteristics of global offshore wind power in the current situation, the main challenges of offshore wind power operation and maintenance market. The innovation of offshore wind power operation and maintenance is discussed from the aspects of operation and maintenance management of offshore wind power, monitoring, analysis technology of units, new wind power operation, maintenance vessels, operation and maintenance robots.

Therefore, this paper is committed to improving the quality of equipment, enhancing operation and maintenance capabilities, reducing risks, saving costs and improving efficiency as the development goals of offshore wind power. By summarizing the existing technologies and related research, the urgent problems to be solved in the development of offshore wind farms are overcome.

## 2. Development and Characteristics of Far-Reaching Sea Breeze Electric Fields

At present, in response to global climate change, major economies have formulated carbon neutrality targets. Carbon emission reduction has driven the accelerated transformation of the global energy structure, moving from the era of fossil energy to the era of renewable energy. The energy revolution has ushered in new development opportunities for the wind power industry. Since Denmark put the world's first offshore wind farm, Vindeby [4], into operation in 1991, offshore wind power has entered a phase of rapid development. Compared with land-based wind power, offshore wind power resources are richer. Offshore wind turbines do not need to occupy land resources, are closer to the electricity load center and have higher utilization hours of power generation [5–8]. Under the background of the rapid development of wind power industry, the development speed of offshore wind power is ahead of the industry.

At present, offshore wind power development are mostly concentrated in offshore waters, facing the problem of competing for limited resources with offshore aquaculture, fishing and transportation routes. Moreover, in the context of energy transformation, with the increase of energy demand, it is an inevitable trend to increase the development of offshore wind power in the open ocean [9]. The development space of offshore wind power in open ocean is huge, which is of positive significance to the technological progress, economic development and energy structure optimization of the industry.

The cost of operation and maintenance of offshore wind turbines for large amounts of wind power mainly includes the operation and maintenance of wind turbines, ship maintenance and insurance. High unit failure rate and high maintenance workload are the biggest difficulties in offshore wind power operation and maintenance. The direction of

development and the cost of leveling electricity is reducing. Turbines with a larger impeller diameter can provide greater annual power generation, thereby reducing costs.

With the rapid development of offshore wind power, the large-scale wind turbines, deep water offshore layout and the use of floating wind turbines pose new challenges to the operation and maintenance of offshore wind power. In the future, smart offshore wind farms need intelligent and efficient operation and maintenance support. The existing data show that under the same installed capacity, the operation and maintenance cost of offshore wind power is twice that of onshore wind power [10], which is more than a quarter of its power consumption cost [11].

The cost of offshore wind power operation and maintenance [12] is shown in Figure 1, including wind turbine operation and maintenance and ship maintenance and insurance. Purchasing prices for offshore wind turbines are falling due to investors' cost considerations and the price war between machine suppliers, which also leads to the use of more cheap components, resulting in a lower overall configuration and difficult quality assurance. High unit failure rates and large maintenance workloads are the biggest difficulties in offshore wind power operation and maintenance.

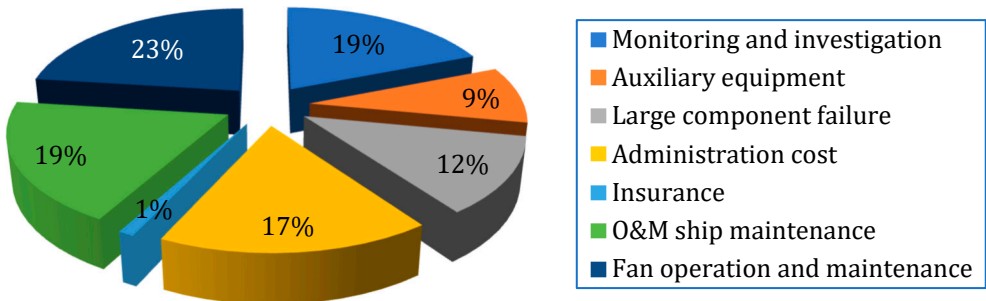

**Figure 1.** Operation and maintenance cost of offshore wind power.

Due to the influence of tide and other conditions, there are window period restriction in the operation and maintenance of offshore wind power, the accessibility of offshore wind turbines is poor. The adverse meteorological conditions and harsh sea conditions limit maintenance time [13,14] and also bring greater security risks. At present, the operation and maintenance mode of offshore wind farms mainly draws lessons from the type of onshore wind farm and adopts three modes: fault maintenance, regular maintenance and condition-based maintenance [15].

1.  Fault maintenance refers to the maintenance after the fault occurs, which is an inevitable maintenance method under the current technical conditions of offshore wind turbines [16]. Fault maintenance requires operation and maintenance personnel to investigate the causes of the fault on the spot, so there are high requirements for weather conditions, ships and spare parts. The maintenance cost and outage loss are related to the fault type and maintenance time, which will affect the reliability of the unit and the power generation income of the whole wind farm.

2.  Periodic maintenance is the preventive inspection and maintenance of wind turbines based on the maintenance plan formulated in advance, mainly for the state inspection and functional test of wind turbine components [17]. Regular maintenance can allow the wind turbine to operate in an optimal state. At the same time, considering the utilization rate of wind resources, regular maintenance is generally arranged for implementation during seasons of low winds. Reasonable time intervals between periodic maintenance are very critical, too large a time interval easily leads to insufficient maintenance of the unit and a decline in reliability, while too small a time interval leads to increased maintenance costs.

3.  Condition-based maintenance refers to the maintenance strategy based on the relevant state information extracted by the wind turbine condition monitoring system and the results of online or offline health diagnosis or fault analysis systems [17]. Its advan-

tage is the combination of the health status of wind turbines, spare parts, weather conditions, etc., and the selection the optimal time point to arrange maintenance in advance to ensure the high availability of wind turbines. However, so far, due to the imperfection of the wind farm data analysis systems and the limitation of the effectiveness of the collected data, offshore wind turbines usually need to be further analyzed in order to confirm the health and fault state of the wind turbine by utilizing manual on-site detection [16].

Based on the above analysis of the literature, the correct development direction has the following three characteristics. First, improving the quality and capacity of newly installed turbines. Second, enhancing the operation and maintenance capacity of the wind turbines, reducing the fault risk and elimination rate. Third, improving the production technology and reducing the manufacturing cost of wind turbines, generator and other parts as well as the operation and maintenance cost after the unit is put into operation. In other words, wind turbines should develop in the direction of large-scale single-machine power, large-scale total installed capacity, intelligent operation and maintenance management and diversified sources.

Most of the improvement directions involved are closely related to the operation and maintenance management of offshore wind turbines. In the process of adopting wind power generation, it is very important to ensure the normal operation of wind turbines, which requires the effective monitoring of the units. However, wind turbines are usually large and contains a lot of parts, and sometimes the quality of the parts is not high. The failure rate under the high intensity workload is high, leading to the decline of the overall reliability of the unit and reducing the actual utilization rate of wind resources [18]. In addition, rich wind resources are often located in the place where is inconvenient traffic, e.g., sparsely populated areas, and huge wind turbines make fault detection and unit disassembly process extremely difficult. This process is particularly difficult for offshore wind power, especially considering the transportation and maintenance costs, making the actual operating costs increase [19], greatly reducing the economic benefits of wind power. The key problems to be solved in the field of offshore wind power operation and maintenance are as follows:

1. Improve the offshore wind turbine operation condition monitoring system, using the unit's health diagnosis technology to identify abnormalities in the unit and predict the unit life.
2. Further optimize the maintenance strategy of offshore wind farm, standardize the operation and maintenance mode, optimize the operation and maintenance resource management scheme and rationally allocate the operation and maintenance resources. The formulation of operation and maintenance strategy should be combined with the reliability data of the unit, so as to improve the efficiency of a single seagoing operation as much as possible in order to avoid frequent seagoing and save operation and maintenance costs.
3. Units with fault-tolerant operation ability can still work smoothly for a certain period of time after certain faults occur. In the case of offshore wind turbines, failure is difficult to avoid, and the fault-tolerant operational function of units has important research value.
4. The correlation of multi-component faults of offshore wind turbines should be studied and the correlation of function and structure between multi-component should be analyzed.

## 3. Monitoring and Operational Maintenance Technology

### 3.1. Offshore Wind Turbine Monitoring and Analysis Technology

The intelligent monitoring of offshore wind power includes underwater intelligent monitoring, structural fatigue and damage monitoring, cable monitoring and foundation scouring monitoring. Intelligent analysis technology covers weather forecasting and early warning systems, window management systems, ship routes, personnel management

systems and maritime security systems. Online monitoring technology transmits and visualizes observation data in real-time through a variety of communication media, which facilitate data processing. With the help of underwater robots, underwater intelligent monitoring can directly monitor the overall state of the foundation by inspecting the underwater part of the target unit. For the fatigue and damage monitoring of the support structure, the deformation, stress, displacement, vibration and corrosion status of the structure are monitored by sensors and the data are transmitted to the monitoring system in real-time. The specific monitoring items and the location of the monitoring points can be selected according to the specific operation and maintenance and safety assessment requirements of the wind farm. Marine environmental parameters monitoring acquires real-time project sea area data through sensors and high-speed transmission technology, including wave data, wind data, current data and temperature and salinity data, and accumulates background parameters for offshore wind power operation and maintenance. The submarine cable monitoring is usually based on the cable online monitoring method of optical fiber and partial discharge, and the operation state of submarine cables is monitored [20]. Sea area monitoring reduces the cable damage caused by ship mooring by continuously monitoring the passing ships around the submarine cable. Foundation scouring monitoring can obtain parameters such as scouring depth through data acquisition equipment, which can be combined with support structure response monitoring to provide higher security for offshore wind power foundation structures.

### 3.2. Operation and Maintenance Management Technology of Offshore Wind Power

Among the life cycle costs of projects in open sea areas, the operation and maintenance costs are second only to the costs of wind turbines. According to statistics, offshore wind farm operation and maintenance costs accounted for 18~23% of the total cost of offshore wind power projects, far higher than the land wind farm operation and maintenance costs, which accounted for 12% of the total cost of the project [16]. According to the estimation of DNV Classification Society, the average number of annual outages of an offshore wind turbine can be as high as 40 times, and the overall probability of failure is maintained at about 3%. On average, about 30 offshore wind turbines need a professional offshore wind turbine maintenance ship for daily maintenance work. Therefore, the demand for offshore wind turbine maintenance ships (see Figure 2) for offshore wind farm supporting services is increasing, and the professional requirements of ships are also increasing. While improving the reliability and safety of operation and maintenance, reducing the cost of operation and maintenance is also a major problem of offshore wind power.

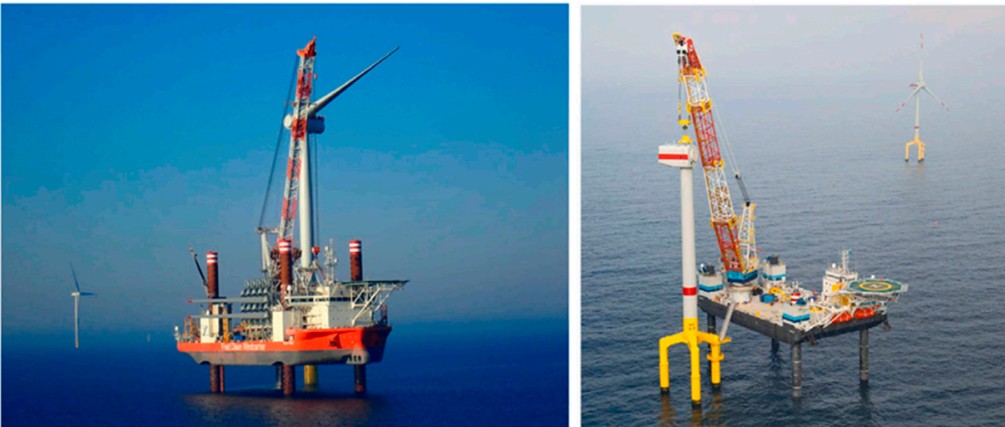

**Figure 2.** Self-elevating and floating operation and maintenance platform [12].

### 3.2.1. Operation and Maintenance Operations

The processes of operation and maintenance are mainly divided into the following three types:

1. Periodic Maintenance

Periodic maintenance refers to the periodic detection and maintenance of wind turbines according to the technical requirements of wind turbine manufacturers and operation time. The workload is relatively fixed, and there are generally relatively standard procedures and requirements [21]. Through regular testing, the equipment can maintain an optimal running state, prolong the service life of the wind turbine, produce more economic efficiency, make full use of resources and maximize benefits.

2. Daily Operation and Maintenance

Daily operation and maintenance work is mainly the disposal of various faults. The prediction, detection and elimination of wind power equipment faults require personnel to have professional skills in electrics and communication [21]. This work is also one of the most technical and challenging tasks for the operation and maintenance of wind turbines. The working experience, technical level and knowledge reserve of the personnel determines the speed and effect of processing, which directly affects the normal operation of wind power [21].

3. Accident Maintenance

When large components of wind turbines are damaged, such as blades, generators and gearboxes, they need to be removed for repair. This failure generally produces a large number of repair costs, and the repair process is more complex. After an accident occurs, experts should be organized to analyze the causes and treatment measures immediately, and a detailed repair plan should be developed and fully evaluated before implementation.

3.2.2. Operation and Maintenance Ship

In regard to offshore wind farm operation, maintenance ships are an important means of transportation for a wind farm to maintain normal construction, operation and maintenance. Common ships used can be divided into four types: ordinary ships, professional ships, mother ships and jack-up ships.

At present, 400 offshore wind power carriers have been put into use globally. The new type of offshore wind power carriers is larger in size and can carry more equipment and components. They have good riding comfort, faster speed, higher safety of personnel transfer and stronger ability to resist wind and waves. There are many types of professional wind power operation and maintenance ships. Their special bows can not only be used for operation and maintenance personnel to climb stairs from the bottom of the wind turbine but also to reduce the ship's shaking. The main types of wind power ship include mono-hull ship, twin-hull ship, three-hull ship, small hydroplane catamaran, surface effect ship and small hydroplane trimaran. The typical parameters of various types of ship are shown in Table 1.

**Table 1.** Typical parameters of different types of ship [7,14].

| Major Parameter | Mono-Hull Ship | Twin-Hull Ship | Three-Hull Ship | Small Hydroplane Catamaran | Surface Effect Ship | Small Hydroplane Trimaran |
|---|---|---|---|---|---|---|
| Master/m | 21 | 20 | 18 | 20 | 28 | 27 |
| Maximum speed/kn | 23 | 25 | 20 | 23 | 33 | 25 |
| Maximum manning | 12 | 12 | 12 | 12 | 12 | 12 |
| Dead weight/t | 5.0 | 10.0 | 1.0 | 2.0 | 4.0 | 4.5 |
| Maximum significant wave height/m | 1.5 | 2.0 | 2.5 | 2.5 | 2.5 | 3.0 |

Operation and maintenance vessels are the main means of transportation for offshore wind farm construction, operation and maintenance, which plays the role of platform carrier. It provides services for the operation and maintenance of wind turbines in offshore wind farms, minimizes and reduces the operation and maintenance time and cost, reduces rotary failure and shutdown, improves power generation efficiency and improves the

economic benefits of wind farms. It can quickly and comfortably reach the wind column site and is reliable and stable, which ensures that people can proceed safely and quickly. The operation and maintenance of transport vessels are shown in Figure 3.

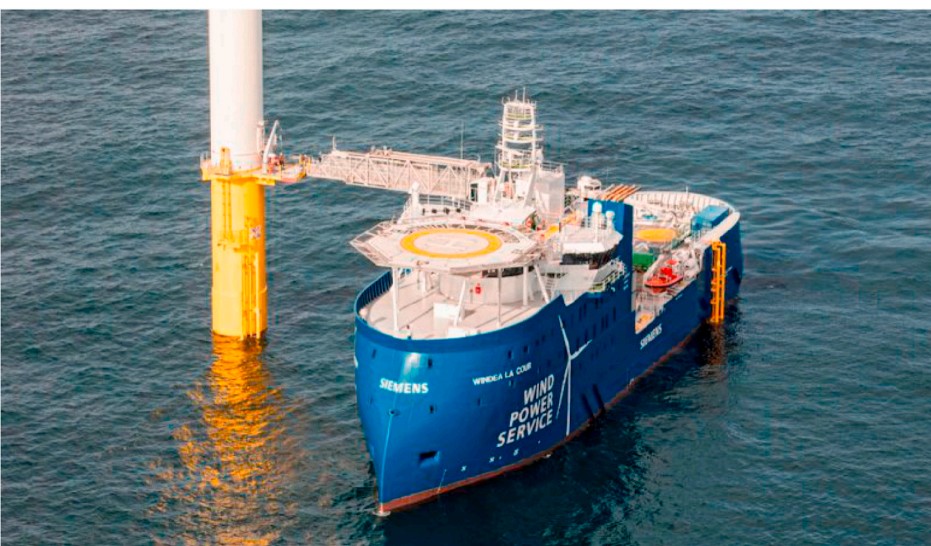

**Figure 3.** Operation and maintenance of traffic vessels [22].

The main performance characteristics of the operation and maintenance of the transportation ship are as follows:

1. Rapidity and flexibility. The ship requires high speed to reduce midway sailing time and reach the wind field quickly.
2. Comfort. The ship requires good seakeeping and low noise to retain the operational effectiveness of personnel.
3. Berthing. Good maneuverability of the ship, excellent equipment and the ability to reach and stabilize the wind power column is necessary.
4. Safety. Operators must be able to safely and smoothly get up and down the wind power columns by means of operation and maintenance of transport vessels.

Large-scale offshore wind fields in Europe mostly use large carriers, which are far away from the shore and have deep operating water depths. This kind of operation and maintenance mother ship generally adopts the catamaran type, which can meet the operation and maintenance of offshore wind power under the harsh sea conditions of level 7 wind, 2.5 m wave height and 2 kN surface velocity. With the development of offshore wind fields in far-out, open sea areas, the demand for large-scale carriers will increase significantly in the next 5~10 years.

### 3.2.3. Operation and Maintenance Platform

The main operation projects of the wind power operation and maintenance platform include the maintenance and replacement of the support tower, cabin and blade of the wind turbine [23]. At sea, whether in the installation of the wind turbine or the foundation or the replacement of large components, the corresponding transportation tools are needed for transportation to the wind field and must be equipped with corresponding equipment to be installed in place. The wind power operation and maintenance platform usually shuttles between several adjacent wind fields to maximize the platform capacity during the late stages of wind field construction. The wind power operation and maintenance platform (see Figure 4) mainly has the following three characteristics:

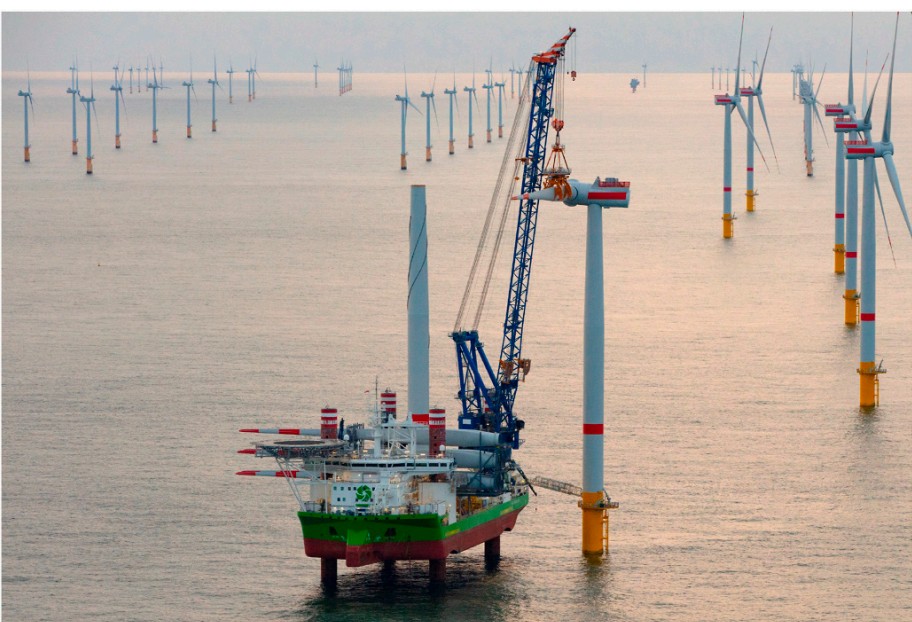

**Figure 4.** Wind power operation and maintenance platform [24].

1.      Fast Lifting Speed

Unlike offshore oil drilling platforms, the offshore operation environment of wind power operation and maintenance platform is complex, the operation time window is short, and the sea conditions are bad. The operation and maintenance platform needs to move the ship station's piles and the platform lifts frequently. Therefore, there are high requirements for the lifting speed, reliability and durability of the lifting system equipped on the platform. At present, the world's most common platform lifting system is mainly divided into a hydraulic plug type and gear rack type. Compared with the hydraulic pin lifting system, the gear rack type has the characteristics of fast speed and easy platform adjustment.

2.      Operation Depth

With the continuous progress of technology, offshore wind power will develop for farther out and deeper sea areas, and the operating water depth could reach 60 m. In the future, offshore wind power will move towards the deep sea. At present, the planned water depth of an offshore wind field reaches 50 m. Therefore, the operating water depth becomes the operating range and the decisive factor of the wind power operation and maintenance platform. At this time, the importance of leg length is highlighted. The leg form of operation and maintenance platform is mainly divided into a cylindrical leg and truss leg. Compared with the cylindrical leg, the truss leg can effectively reduce the force of waves, currents and sea wind on the platform due to its hollow structure, which greatly enhances the stability and safety of the platform. At the same time, due to its own structural characteristics, the cylindrical leg is generally suitable for platforms with operating water depths of not more than 40 m, while the truss leg can meet the deeper operating water depths.

3.      Self-Propelled and Dynamic Positioning Capability

The operation and maintenance platform has the ability of self-navigation, which can realize the rapid ship relocation between wind farms and towers. Compared with the tugboat towing and mooring positioning, it is more accurate and faster, which greatly saves time at the location and the cost of tugboat anchor and effectively improves the operating efficiency of the platform [25]. A platform equipped with a dynamic positioning system (DP) can automatically maintain its position and heading under the specified environmental conditions. At the same time, it also has an independent centralized manual position control and automatic heading control. Moreover, the dynamic positioning system

can maintain the floating positioning of the platform due to its ability to resist waves and currents, which has significantly enhanced the security and reliability of the self-sustaining platform during storms.

### 3.2.4. Wave Compensation Ladder

It is well known that the wind field is characterized by large waves and numerous dark currents, and it is very dangerous for personnel move up and down a tower. How to ensure the safety of personnel up and down the tower is very important. The traditional method for personnel to ascend and descend a wind tower was by using a ship to reach the wind tower in order to allow personnel to seize the opportunity to climb the wind tower. This method is very dangerous, as a little carelessness could cause a tragedy, as, after all, wind and sea conditions are bad, the environment is complex and changeable, and the ship is difficult to maintain in a relatively stable state for a long time [26]. Ordinary port ladders are widely used in marine engineering but due to the limitations of their placement along the port, they can only be applied to the boarding of persons between ships or platforms and boats or docks. The common footbridge is also a kind of personnel boarding device commonly used in marine engineering. However, due to its design limitations, the wave is a little too large to be used. The wave-compensated starboard ladder boarding system can safely transport the operator to the wind tower platform under the conditions of grade 4 sea. It has automatic control ability in the complex and changeable sea environment and can maintain relative balance.

The wave compensation starboard ladder mainly includes the six-degree-of-freedom compensation ability of the compensation mechanism, the telescopic ability and pitching ability of the telescopic starboard ladder and the bearing capacity of the starboard ladder. The basic part of the compensation structure is a six-degree-of-freedom self-balancing compensation platform, which realizes integration and modularization, and can compensate for the relative displacement caused by waves. The special wave compensation ladder with high stability requirements is installed to form a safety channel between the ship and the wind turbine column, which is convenient for personnel safety and ensures personnel safety. In addition, the wave compensation ladder has scalability, which takes up only a small space when it is not being used for collection, which saves deck space, and can also be used for stacking with other items.

### 3.2.5. Intelligent Operation and Maintenance Robot

The harsh marine environment brings a lot of inconvenience to the operation and maintenance of offshore wind power. In order to improve the convenience of operation and maintenance and to reduce the safety risks to operation and maintenance personnel, robots and unmanned aerial vehicles are applied. Robot systems is also a key part of the intelligent operation and maintenance of offshore wind power. Figure 5 shows the robot 'BladeBUG', which is designed for the maintenance of offshore wind turbine blades. The robot has a crawler and six crawling feet. The end of each foot has a vacuum bonding device, which can be firmly adsorbed on the surface of the blade and carry out flexible crawling. Using robots instead of manpower for operation and maintenance can reduce the risk of operation and maintenance accidents, improve detection efficiency and accuracy and save maintenance costs [27,28]. It is estimated that a good robot system in the future could help wind power projects save USD 33 million throughout their life cycle. The first European offshore wind power operation and maintenance robot test center was established in Portugal in 2020, which is dedicated to research on robot operation and maintenance operations in harsh environments.

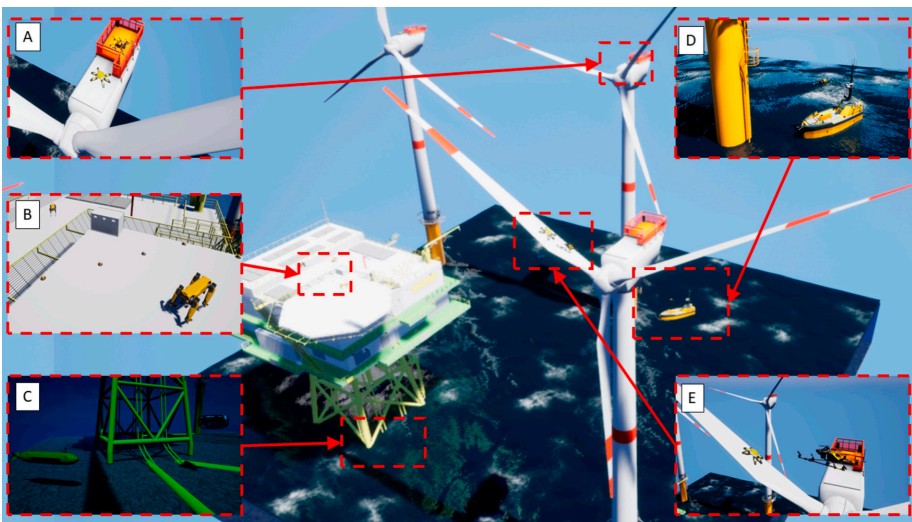

**Figure 5.** Intelligent operation and maintenance robot [29].

### 3.2.6. Limitations and Future Applications

At present, the operation and maintenance operation of offshore wind farms generally includes preventive maintenance, daily maintenance and post-repair maintenance. Both methods include limitations: the cause of the fault of the unit and the faulty parts, making the relevant preparation impossible and increasing the time consuming and loss of maintenance [30]. In addition, the current operation and maintenance processes are mainly for the operation and maintenance of ships and operation and maintenance platforms, and it is difficult for new technologies to display their roles. Through the above review, we need to focus the two main methods that the monitoring, operation and maintenance technology can utilize to solve the development problems of offshore wind turbine operation and maintenance management. Firstly, it is committed to the real-time monitoring of the information generated during the operation of offshore wind turbines and maintenance in abnormal modes that may lead to failure in order to reduce the operation and maintenance costs and risks. Secondly, through the comprehensive analysis of wind farms, the appropriate operation and maintenance methods are selected or combined, and the deployment of different operation, maintenance equipment and the efficient application of new technologies are managed through operation and research planning.

## 4. Deep Sea Breeze Electric Field Monitoring and Operational Risk

The intelligentization of the whole life cycle of offshore wind power is the key to achieving the optimal power cost of offshore wind power leveling, and the intelligent operation and maintenance of offshore wind power is a systematic project.

Intelligent operation and maintenance systems use big data and intelligent data technology to make operation and maintenance decisions based on data. Using fine cost control, through the real-time calculation of turbine operation and maintenance cost and income throughout the whole life cycle, the fine level of operation and maintenance management are improved and the operation and maintenance costs are effectively reduced. This will perform well in offshore wind turbine component level tests, building offshore 'genetic engineering' wind power, formulating reasonable operation and maintenance plan and improving the reliability of unit operation. According to the early warning information of large parts, locking the lifting vessels and spare parts of large parts in advance shortens the shutdown time of large parts [31]. Through the integrated system of fault warning and operation inspection and maintenance, the operation and maintenance plan and schedules are optimized to reduce the operation and maintenance cost of offshore wind power. The performance of offshore wind power generation and the cause of power loss can be evaluated, and the research and application of control strategy optimization technology can be

carried out to further improve the power generation of wind turbines, optimize regional wind transportation tools in different sea areas and on different scales, improve operation and maintenance efficiency and reduce daily operation and maintenance traffic costs.

It is the development goal of offshore wind power operation and maintenance mode to scientifically and reasonably plan operation and maintenance time and route, adopt pre-operation and maintenance modes to eliminate hidden trouble and reduce operation and maintenance costs. For the planning of the operation and maintenance path, the safest, most convenient and most cost-optimal operation and maintenance transportation route should be selected based on the wind power prediction, the accessibility of operation and maintenance ships, the operation status and the health status of the units [32].

The operation and maintenance risks of offshore wind power construction projects are divided into the following parts.

1. Equipment Failure Risk

Equipment failure is a high risk of offshore wind power construction project in the operation stage. Failures include blade control system faults, transformer faults, generator or gearbox faults, offshore booster station electrical equipment faults, HVAC equipment faults and fire equipment faults. Monitoring and video equipment failures, submarine cable failures and equipment failures bring uncertainty risks to the normal operation of offshore wind farms.

2. Personnel Safety Risk

During the operation and maintenance period, the maintenance personnel frequently travel between the offshore wind field and the land. The personnel board the wind turbine platform at sea and maintain operations on the wind turbine platform, and repair operations at high places inevitably produce personnel safety risks. There are personnel safety risks such as electric shocks, falling from high places, personnel falling into water and equipment damage.

3. Maritime Traffic Risk

The following two means of transport (operation and maintenance) are often used in offshore wind farms: one carrier and two helicopters. These two types of transportation are affected by sea conditions and weather. The operation and maintenance vessels are the main commuting tool for the operation and maintenance of offshore wind power. At present, ordinary transport vessels are still used as the main transportation tool, which have the disadvantages of poor wave resistance and poor berthing ability. It is difficult to meet the requirements of safe navigation, such as wind and wave resistance, anti-collision and maritime rescue.

The risk-based maintenance method can reduce the overall maintenance difficulty and life cycle cost and present an availability and unit performance satisfactory to all parties. The traditional operation and maintenance strategy of planned maintenance combined with fault maintenance consumes a lot of manpower, material and financial resources. With the continuous progress of operation and maintenance technology, the operation and maintenance strategy based on condition-based maintenance has become a developing trend [14].

Condition-based maintenance takes the equipment status as the starting point, finds the latent faults through online monitoring and offline measurement and evaluates the equipment status. Condition-based maintenance is highly targeted. Through the comprehensive analysis of equipment, it is better to determine whether the equipment is to be overhauled and the maintenance effect is also better [33]. The operation and maintenance strategy based on condition-based maintenance can facilitate the unified scheduling of operation and maintenance resources for the maintenance of multiple units, improve the efficiency of single outbound operation, reduce the number of outbound operations and reduce transport costs [34]. Condition-based maintenance is a major innovation in the operation and maintenance management of offshore wind power. The realization of

condition-based maintenance requires the monitoring and analysis of the operating state of the unit and the strengthening of the life cycle, the monitoring of components and the condition monitoring of large components in combination with the characteristics of different offshore wind farms.

The research related to maintenance status is extensive, there are many documents to demonstrate maintenance methods. At present, many complete maintenance processes have been formed in the literature, and the optimization schemes are given through empirical and operational research decisions. For example, the research of Ren Y [35] et al. (2015) proposed the process of wind turbine status maintenance, including data collection, online monitoring, fault diagnosis, fault prediction, decision-making and status maintenance implementation. A study by Igba J [36] et al. (2019) demonstrated the introduction of CBM technology and its application method based on CM technology system, which arranged maintenance plans according to the status information of equipment to prevent major faults for a series of large devices with high operating costs and difficult maintenance. The research also introduced PHM integration technology, which is mainly used to reduce maintenance cost and improve maintenance decisions, providing feedback from product use for product design and verification processes.

To sum up, the related technologies of data monitoring and acquisition in maintenance processes have become mature, and there are many mature research methods to integrate data acquisition into the maintenance process. At present, papers on reducing operation and maintenance risk and status are mainly divided into operation methods of maintenance and related models of risk and abnormal data processing. The combination of the two needs to be urgently developed.

## 5. Smart Wind Farm Technology

### 5.1. Definition of Smart Offshore Wind Farm

Smart offshore wind farm refers to a new offshore wind farm, which is widely used in cloud computing, big data, Internet of Things communication, artificial intelligence and other new technologies.

### 5.2. Construction of Intelligent Operation and Maintenance Management Platform for Offshore Wind Farms

The intelligent operation and maintenance of offshore wind farms in open sea areas aims to achieve the lowest cost, high reliability and increase the power generation of wind power operation and maintenance and arrange the maintenance strategy, business process, time cycle, transportation tools and maintenance personnel of offshore wind farms through intelligent algorithms [37]. By building an open sea intelligent operation and maintenance management platform, the remote and centralized control mode of the wind farm is realized, the reliability and availability of wind turbines are improved and the economic benefits of the wind farm are ensured.

The offshore wind farm intelligent operation and maintenance management platform includes wind farm intelligent monitoring and operation optimization systems, wind farm integrated information management systems, wind farm operation and maintenance scheduling management systems and wind turbine condition monitoring and health management system.

5.2.1. Wind Farm Intelligent Monitoring and Operation Optimization System

The intelligent monitoring and operation optimization system of the wind farm is a system that integrates wind turbine monitoring, power dispatching, meteorological information and operation optimization based on big data. It enables wind farm managers to understand the operation status, fault information, meteorological environment and power dispatching information of wind turbines in good time and plays a guiding role in the operation and maintenance decision of wind turbines.

1.  Meteorological information, such as sea weather, typhoons, rainstorms, snow and disaster warnings, are analyzed and recorded. According to the meteorological information, the early warning and pre-processing schemes and the sea operation strategy are arranged.
2.  Receiving and executing the instructions of the local power dispatching department, participating in the voltage regulation and frequency regulation of the power grid and peak and valley filling. At the same time, with the continuous development of the new power market, wind farms in far-out sea areas can also participate in power transactions to further meet the demand for load characteristics.
3.  Condition monitoring, early warning and fault display of all equipment in the wind farm and the embedded application of fault treatment scheme based on the expert database can improve the informatization and intelligence level of the wind farm; at the same time, it carries intelligent data and information collection strategy to provide basic support for wind farm operation optimization.

For example, research by Ouadie Bennouna et al. [38] discussed principal components analysis (PCA) for faults detection in an offshore wind turbine generator (OWTG). An accurate analytic modeling of healthy and faulted OWTG was suggested to achieve the data matrix needed for PCA method.

### 5.2.2. Wind Farm Integrated Information System

The wind farm integrated information management system includes at least four databases [39]: equipment archives, equipment information database, warehouse and component replacement maintenance management database, inspection and fault work order database, etc.

1.  The equipment archives record the basic information of various types of equipment of wind turbines, such as wind turbine type, configuration, component type, ship information, etc., which is convenient for subsequent equipment classification statistics, device quality tracking, operation scheduling management, etc.
2.  The object of the equipment information database contains the technical specifications, operation and maintenance manuals, spare parts, tools and transportation tools involved in the maintenance process of wind turbines. Ships are the main means of transportation, and the basic information of ships and the seagoing operations are recorded. If the ship is temporarily leased, it is necessary to record the relevant information of leasing.
3.  The warehouse and spare parts replacement and maintenance management library is used to record the quantity, quality, storage location, access and other basic information of spare parts and tools. At the same time, the relevant information before and after spare parts replacement and the whole process of spare parts self-maintenance or third-party maintenance are recorded in detail, which is convenient for subsequent inquiry and statistical analysis.
4.  The inspection and fault work order database is used to record wind turbine fault information, maintenance information, etc. The fault work order information involves the historical faults of the unit, the fault treatment method, the time and type of the replacement parts, etc., and these experiences are used to establish and improve the expert database. Maintenance information records and tracks the whole process of maintenance according to the requirements of plan management, including requirements, plans, processes and reports.

For example, Haitao Wang et al. [40] proposed to use the blockchain to build a wind farm information system to solve these problems. This paper analyzes the applicability of blockchain and wind farm information systems and designs the functional modules of the wind farm information system and the corresponding system framework.

### 5.2.3. Wind Farm Operation and Maintenance Scheduling Management System

Due to the constraints of marine factors, wind power in far-out sea areas cannot adopt the mode of real-time maintenance of land-based wind power. Instead, it is necessary to select appropriate maintenance strategies based on a comprehensive analysis of offshore maintenance costs, outage power loss and marine climate conditions. Wind farm operation and maintenance scheduling management system includes planning scheduling management and trigger scheduling management.

The planning scheduling management mainly aims at the various maintenance requirements and inspection work of wind farms. Taking the maintenance of a single unit as the task, the operation items, tools, spare parts, and ship requirements are listed. After the maintenance, the maintenance work is summarized, including the evaluation of the quality of the maintenance work.

### 5.2.4. Condition Monitoring and Health Management System of Wind Turbine

The structures of offshore wind turbines and onshore wind turbines are basically the same, and the common faults are also concentrated on several key components, such as blades, gearboxes and generators [41]. The condition monitoring and evaluation system of wind turbine mainly monitors the status of blades, spindles, gearboxes, generators and electrical system components and evaluates the health status of each component, such as faults or hidden dangers. Early warning and reasonable maintenance suggestions are given to prolong the service life of components. A study by Volanthen M et al. [42] provides a method for monitoring wind turbine performance, which uses bending moment data from strain sensors in the wind turbine blades to calculate the rotational speed of the wind turbine, the angular position of the turbine blade and the drive torque. As a result of the load on the rotor, the advantage of this method is that the input to the wind turbine drive train can be measured directly.

The condition monitoring and health management system of wind turbines are associated with the integrated information management system of wind farms and the operation and maintenance scheduling management system of wind farms. The maintenance personnel call the operation and maintenance scheduling management system of wind farms according to the prediction results of the condition monitoring and health management system, and the plan is included in the plan management library. At the same time, the system is also associated with the equipment unit archives, equipment information library, warehouse and component replacement maintenance management library, environmental weather information, etc., which can be included in arrangements for maintenance time, maintenance cycles, transportation and maintenance personnel. See Table 2 for a comparison of different systems.

**Table 2.** The advantages and disadvantages of the smart operation and maintenance management platform for wind farms [43].

| | Wind Farm Intelligent Monitoring and Operation Optimization System | Wind Farm Integrated Information System | Wind Farm Operation and Maintenance Scheduling Management System | Condition Monitoring and Health Management System of Wind Turbine |
|---|---|---|---|---|
| Advantage | Real-time status monitoring | Information integration processing | Consider environmental factors | Real-time health monitoring |
| Disadvantage | Troubleshooting of type I and II errors is prone to occur | Operating system penetration is low | Integration with elements needs to be improved | Troubleshooting of type I and II errors is prone to occur |

### 5.3. Application Analysis of 5G Technology in Operation and Maintenance of Smart Wind Field

At present, the mobile communication technology represented by 5G is closely integrated with new technologies, such as artificial intelligence, Internet of Things, cloud computing, big data and edge computing, providing more and more applications and opening a new era of interconnection. The 5G network changes the operation mode and

operation mode of the core business of the energy industry with ultra-high bandwidth, ultra-low delay [1] and ultra-large-scale connection technology, and comprehensively improves the operation efficiency and intelligent decision-making level of the traditional energy industry.

According to the investigation and study, several offshore wind power projects have successfully applied mobile communication networks, such as China's Jiangsu Rudonghaishang wind farm and Dafeng offshore wind farm, which have built 4G mobile communication network. However, the 5G mobile communication network has not been successfully applied to offshore wind power projects.

At present, the operation and maintenance mode of offshore wind farms mainly draws on the operation and maintenance mode of onshore wind power. Artificial or local operation and maintenance is still the main means of current operation and maintenance work, which has multiple problems, such as long operation and maintenance cycles and low operation and maintenance efficiency. In order to achieve intelligent, accurate and machine operation and maintenance work, the application of advanced technology is the key. According to the characteristics of offshore wind power, the following typical application scenarios of 5G technology will be the main trend in the intelligent operation and maintenance of offshore wind farms in the future [44].

### 5.3.1. 5G Unmanned Aerial Vehicles and Robots

The eMBB and mMTC technologies in 5G can be applied to endow the UAV with important capabilities such as real-time ultra-high-definition image transmission and remote low-delay control to realize the fine multi-directional inspection of offshore wind power and achieve the 'unattended, less on duty' operation requirements of offshore wind power.

The inspection robot of the booster station is equipped with intelligent infrared imaging equipment, laser positioning and navigation, multi-axis universal arm, adaptive wheel chassis, intelligent charging pile and other intelligent components. By using the 5G network and big data processing platforms, the operation parameters, such as the temperature of electrical equipment of booster station, the information of the automatic control device and the switch position state, are regularly cruised, collected, analyzed and warnings are given, and the on-site emergency disposal is carried out under the remote control of the on-duty personnel.

### 5.3.2. Tower Inspection and Real-Time Feedback

In the traditional inspection of wind turbine towers and other parts, ground telescope visual inspection or high-altitude hanging basket manual inspection are mostly used. The inspection operation process is complex, and the safety hazards are great. Since the tower drums of units are mostly made of metal magnetic conductive materials, the magnetic adsorption wall-climbing robot can be considered for use in inspections [45]. With the continuous development of industrial internet technology combined with 5G technology, robot inspection can achieve remote flaw detection and real-time feedback communication, tower surface fouling removal [46] and tower surface damage repair. A robot computer-aided design (CAD) model of the mechanical mechanism, force and structural analysis and the testing of the prototype model are addressed in the paper by JuiHung Liu [47]. The design utilizes 5G technology to inspect and provide real-time feedback on the tower.

### 5.3.3. Fault Identification and Real-Time Feedback of UAV Blade

The blade of a wind turbine is far from the ground, and it is difficult to effectively identify a blade fault by conventional means. Due to poor accessibility and short window windows for repair in offshore windfarms, it is more difficult to identify the blade fault. The traditional manual repair method causes the turbine to stop for a long time and leads to serious power loss. The UAV can move synchronously with the blade of the wind turbine, without the need for the blade to remain static. UAV blade fault identification has the

characteristics of safety, reliability and efficiency, which eliminates the safety hazards of personnel falling into the sea and objects falling from high altitude. UAV equipped with high-definition imaging equipment can obtain the image of the blade from all directions and angles, which makes the recognition results more reliable [48].

### 5.3.4. Cabin Environment Perception and Internal Real-Time Monitoring

The environment of offshore wind turbines is often accompanied by harsh conditions, such as high temperatures, high humidity and high salinity. When the gearbox, generator and other components in the cabin are running, a large amount of heat is also generated. Combining the above two factors, the temperature and humidity inside the cabin reach a relatively high level, which has a certain adverse effect on the operation of the unit. By monitoring the real-time change of temperature and humidity in the cabin, the temperature field distribution diagram in the cabin can be established and the components that are prone to overheating in the cabin can be found. Corresponding measures are taken in operation and maintenance to ensure the safety of the cabin and the normal operation of the unit [49].

The existing wind turbine engine room monitoring system mainly collects real-time parameters of wind turbine operation, such as temperature, humidity, wind direction and wind speed. Operators cannot directly and comprehensively monitor the operation status of equipment. With the continuous application of high-speed transmission technologies, such as 5G, problems like network bandwidth and transmission speed are solved, and the realization of real-time monitoring in the cabin becomes possible. The real-time monitoring inside the engine room can be used to monitor the working state of the components inside the engine room. The combination of real-time monitoring images and operating parameters helps operators to better monitor the operation of the equipment. In daily maintenance, it is possible to supervise the standardized operation of field operators through the real-time monitoring of images and provide corresponding remote guidance.

### 5.4. Comment on the Development of Intelligent Wind Field Technology

Currently, there is still a big gap between offshore wind farms and digital smart wind farms, mainly reflected in the following aspects: (1) lack of digital intelligent management function in wind field site selection, unit selection and infrastructure process information; (2) lack of intelligent evaluation systems and methods for the operation and maintenance process, quality, operation and maintenance safety supervision; (3) the assessment of the health status of each system and equipment of the wind farm also requires manual participation and no real intelligent assessment is achieved; (4) although the initially integrated information management platform has achieved the collaboration between some production, management and operation businesses, there are, however, still some information islands, such as status monitoring (CMS) data and offline detection data; (5) insufficient intelligent decision-making ability.

Combined with the above related techniques and management methods, in the construction process of a smart wind farm, the following problems should be solved: (1) completing the standardization of all kinds of data and information, unified data interface; (2) completing the digitalization and intelligence of equipment status, personnel assessment, unit performance evaluation and optimization; (3) based on digitalization and information technology, the use of various kinds of intelligent decision-making models, realizing the intelligence of wind farm operation and maintenance decisions; (4) completing the connection between the information intelligent analysis system and the wind farm master control system. In order to enable the main control system to adjust the operating state of the turbine according to the feedback of the information intelligent analysis system.

### 6. Conclusions

The operation and maintenance mode of offshore wind power is developing towards digitalization, intelligence and precision. The intelligent operation and maintenance of

offshore wind power aims to reduce the operation and maintenance cost and improve the power generation efficiency in the whole life cycle of the project. This paper introduces the current offshore wind power monitoring, operation and maintenance technology, builds a smart operation and maintenance management platform for wind farms in far-out sea areas, and draws the following conclusions:

1. Offshore wind power operation and maintenance camp should improve equipment quality, enhance operation and maintenance capacity, reduce risks, save costs and improve efficiency. The improvement of far-reaching maritime maintenance modes and equipment capacity should be the focus of the next phase of wind power operation and maintenance. More far-reaching maritime maintenance methods and equipment will emerge to maximize operational efficiency.

2. The rational use of operation and maintenance ships, operation and maintenance platforms and intelligent robots can achieve cost reduction and the efficiency of offshore wind power operation and maintenance and improve the development capacity and level of wind power industry. While developing monitoring technology, operation and maintenance operations and equipment, it is also necessary to consider the optimization and iteration of the operation and maintenance management mode and the reasonable choice of operation and maintenance equipment.

3. It is necessary to build intelligent offshore wind farms and achieve the intelligent monitoring and management of offshore wind farms. They are of great significance in improving the automation level, operation and maintenance efficiency of wind farms, reducing operation and maintenance costs, improving the economic and social benefits of offshore wind farms and improving the ability to resist risks. This paper analyzes the characteristics of digitally smart wind farms, summarizes the problems of equipment, data, technology and management encountered in the construction process of smart wind farm and provides solutions.

4. Offshore wind farms have a wide range of operations, few personnel and severe natural environments. The intelligent operation and maintenance technology of offshore wind farms based on 5G and other technologies can reduce the operation intensity of operation and maintenance personnel of wind farms, improve the safety and reliability of equipment, effectively improve the management efficiency and management level of wind farms and realize the digitization of operation and maintenance experience, which is an effective technical path to improving the core competitiveness of offshore wind power projects.

**Author Contributions:** Conceptualization, Y.G. and H.W.; methodology, Z.W.; validation, Z.W. and Y.G.; resources, Y.G.; data curation, Z.W.; writing original draft preparation, Z.W.; writing review and editing, Z.W.; visualization, Z.W.; supervision, H.W. All authors have read and agreed to the published version of the manuscript.

**Funding:** This research was funded by the National Science Foundation of China, grant no. 51909188.

**Institutional Review Board Statement:** Not applicable.

**Informed Consent Statement:** Not applicable.

**Data Availability Statement:** The data are from references, which have been cited.

**Conflicts of Interest:** The authors declare no conflict of interest.

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
