# Peer review of "Review on Monitoring and Operation-Maintenance Technology of Far-Reaching Sea Smart Wind Farms"

_jmse, doi:10.3390/jmse10060820_

Round 1
Reviewer 1 Report
The concept behind the manuscript “Research on Monitoring and Operation and Maintenance Technology of Far-Reaching Sea Smart Wind Farm” is interesting.
However, the aims and scope of the paper and the underlying study is not clear. The manuscript indicates that it is a research paper. However, I was unable to find methods, results or discussion related to the research. All the material in the paper appears to be from previous literature, which would suggest a review paper. Even considering the manuscript a review paper it is very general and superficial, appearing to be more an introductory chapter for a textbook on offshore wind farms maintenance. The scientific contribution and novelty of this paper is unclear.
I have some specific comments:
The scope of the manuscript appears to be too general, covering a large number of topics without going in detail on any of them. The novelty and scientific contribution of the paper are therefore unclear. I would suggest clearly defining the scientific contribution and novelty of the manuscript and afterwards developing the narrative and content of your manuscript. It is very important to define if this manuscript is a research paper or a review paper.
Many of the ideas and concepts discussed on the manuscript do not have proper citation or references. Section 1 (introduction) only has one citation, and the second citation only appears until line 82. All the information, ideas and concepts that were not generated by research from the authors needs to be properly cited.
Was figure 1 developed by the authors from data obtained from previous literature or is the pie chart taken from other literature. This needs to be clarified. If the authors developed the pie chart the citation should be in the preceding paragraph for the figure, not in the figure description.
Many of the figures contain photographs and have references. Were the photos taken directly by the authors or obtained from previous literature? Do the authors have copyright authorization to use the pictures and charts in this manuscript? This needs to be clarified. Photographs cannot be included without proper copyright authorization.
The grammar and English structure of the manuscript needs to be reviewed and edited by an English native speaker after the final version of the manuscript is completed. There are a significant number of corrections that need to be included. Many sentences are very long, more than 50 – 60 words on each sentence. Sentences should not exceed 30-35 words. If you have longer sentences, they need to be split into two or three sentences.
Author Response
Thank you for your comments, each of which is of great value to improve our manu-script. According to your suggestions, we have revised the manuscript in detail. Please refer to the attachment for the amendments.

Reviewer 2 Report
The article is of potential interest, please address the following comments:
I will suggest you clarify the contribution if the article aims to provide an overview, such as a review article, a more exhaustive literature review is expected if a particular scientific contribution is presented, please explain it. It seems such as introductory book chapters about the topic without a contribution of archival value, even the more interesting things such as photos and graphics, seem to come from other articles, they are cited, but I don't know if they have copyrights.
Author Response

(The authors gave the same response as above.)

Reviewer 3 Report
The paper under review presents the current offshore wind power monitoring, operation and maintenance technology, and builds a smart operation and maintenance management platform for wind farms in far-reaching sea areas. This reviewer has the following observations:
1) The contributions of the paper should be clearly stated.
2) How the typical parameters of the Table 1 occur?
3) In Figure 6, please check the writing of the references.
4) In each subsection of Section 5, the authors should provide a presentation of examples used in practice using some references.
5) It is suggested to summarize in a table the advantages and disadvantages of the smart operation and maintenance management platform for wind farms in far-reaching sea areas.
Author Response

(The authors gave the same response as above.)

Round 2
Reviewer 1 Report
Many of the review comments were addressed. I would suggest improvement in the following sections.
Many statements in the manuscript still lack reference. I would suggest adding references to the relevant information taken from previous published papers.
I would suggest making sure that all figures and images have the proper copyright authorization to be included in this review paper.
Author Response
Thank you for your guidance on the paper, which has been revised according to the reviewers' comments. Please see the attachment.

Reviewer 2 Report
My comments have been addressed, I recommend acceptance.
Author Response

(The authors gave the same response as above.)

Reviewer 3 Report
The revised paper has been modified according to reviewer’s requests. This reviewer has no other questions.
Author Response

(The authors gave the same response as above.)
